# APOBEC1-Dependent RNA Eiting of TNF Signaling Orchestrates Ileal Villus Morphogenesis in Pigs: Integrative Transcriptomic and Editomic Insights

**DOI:** 10.3390/ani15162419

**Published:** 2025-08-18

**Authors:** Wangchang Li, Wenxin Chen, Yancan Wang, Qianqian Wang, Huansheng Yang, Qiye Wang, Bin Wang

**Affiliations:** 1Hunan Provincial Key Laboratory of Animal Intestinal Function and Regulation, Hunan International Joint Laboratory of Animal Intestinal Ecology and Health, Laboratory of Animal Nutrition and Human Health, College of Life Sciences, Hunan Normal University, Changsha 410081, China; liwangchang@st.gxu.edu.cn (W.L.); yhs@hunnu.edu.cn (H.Y.); 2Yuelushan Laboratory, Changsha 410128, China

**Keywords:** RNA editing, TNF signaling pathway, villus height, ileum, pig

## Abstract

Ileal villus height impacts nutrient absorption. This study stratified crossbred pigs into high and low-villus groups (n = 4) and identified differentially edited genes (REGs) and expressed genes (DEGs) enriched in TNF/IL-17 pathways. Integrative omics revealed that pro-inflammatory genes CXCL10, CCL2, CREB3L2, and PIK3R1 were upregulated but hypomethylated in the low-villus group, highlighting RNA-editing-mediated inflammation regulation of villus development.

## 1. Introduction

Intestinal health is closely linked to nutrient absorption efficiency and overall animal well-being, with intestinal villus height serving as a key indicator of gut health [1,2,3]. While existing studies have primarily focused on enhancing villus height through exogenous nutritional interventions (e.g., probiotics and short-chain fatty acids), the endogenous regulatory mechanisms underlying villus development during animal growth remain poorly understood [4,5].

In formulating management and nutritional strategies to maximize growth performance and health in pigs, it is critical to consider the impact of inflammation on intestinal function [6]. In swine production, pigs are frequently exposed to both pathogenic (e.g., Lawsonia intracellularis) and non-pathogenic stressors (e.g., mycotoxins and environmental changes), which activate the intestinal immune system [6,7]. This activation leads to the recruitment of specialized immune cells and the release of pro-inflammatory cytokines, including tumor necrosis factor α (TNF-α) [8], interleukin 1β (IL-1β) [9], and IL-6 [10]. Excessive production of these cytokines exacerbates mucosal damage by disrupting epithelial integrity (e.g., villus atrophy and tight junction degradation) and inhibiting intestinal development through downstream signaling pathways such as NF-κB and MAPK [11]. Research has highlighted the TNF signaling pathway as a central driver of intestinal inflammation, with its aberrant activation directly linked to reduced villus height and impaired nutrient absorption [12,13]. Therefore, investigating RNA regulatory mechanisms between high- and low-villus phenotypes, particularly focusing on dynamic changes in key genes within the TNF pathway, will provide critical insights for developing targeted nutritional interventions to enhance intestinal health.

RNA editing is a biological process that modulates gene expression and protein function by chemically modifying post-transcriptional RNA molecules [14]. This process not only alters the RNA sequence itself but also indirectly affects protein folding and functional properties through changes in translation. RNA editing is not random; it is precisely catalyzed by specific editing enzymes [15]. The two most extensively studied types are adenosine-to-inosine (A-to-I) editing and cytidine-to-uridine (C-to-U) editing. A-to-I editing is mediated by the ADAR (adenosine deaminase acting on RNA) family. ADAR enzymes recognize adenosine (A) residues in double-stranded RNA and convert them to inosine (I). Since inosine is interpreted as guanosine (G) during translation, this conversion can alter RNA secondary structures, splicing patterns, and amino acid sequences of encoded proteins, thereby influencing critical physiological processes such as neurotransmitter synthesis and immune homeostasis [16,17]. C-to-U editing is primarily catalyzed by the APOBEC (apolipoprotein B mRNA editing enzyme, catalytic polypeptide) family. For example, APOBEC-1 converts cytidine (C) to uridine (U) in apolipoprotein B (ApoB) mRNA, generating a premature stop codon (CAA→UAA) and producing truncated proteins (e.g., ApoB-48 vs. ApoB-100), which regulate lipid metabolism [18,19]. Despite its potential in disease therapy (e.g., genetic disorders and cancer) and developmental regulation, RNA editing research faces two major challenges: lack of standardized quantification, with current methods struggling to accurately measure editing efficiency, limiting cross-study comparability [20], and functional annotation gaps as the biological significance of editing sites (e.g., impacts on protein stability or signaling pathways) remains incompletely understood [21].

To investigate the RNA regulatory mechanisms underlying differences in villus height, this study performed comprehensive transcriptomic analysis on ileal samples from crossbred pigs (Landrace × Yorkshire × Duroc, DLY) (high-villus group, N = 4; low-villus group, N = 4) collected by our research team. High-throughput RNA sequencing (RNA-seq) combined with the RNA editing detection (RE-seq) computational framework was applied to systematically quantify editing events and conduct differential analysis across large-scale RNA-seq datasets [22]. This study provides novel insights into the role of RNA editing in inflammation-related pathways associated with ileal villus height variation, offering a new perspective for intestinal development and health regulation. As illustrated in Figure 1, the research framework highlights RNA editing as a promising target for future studies on gut health, paving the way for the development of RNA-editing-based modulators and their biotechnological applications.

## 2. Materials and Methods

### 2.1. Ethics Statement

All animal experimental procedures were performed according to protocols approved by the animal care advisory committee of hunan normal university (Approval number: 2023-209, approval date: 1 January 2023), Changsha, Hunan, China.

### 2.2. Experimental Animals and Data Storage

This study utilized 90 genetically identical crossbred pigs (Duroc × Landrace × Yorkshire) subjected to a standardized feeding regimen over 100 days. At the end of the trial, piglets were euthanized, and ileal tissues were collected following standardized dissection protocols and labeled with unique sample codes. The ileal samples were processed as follows:

Histopathological analysis: Portions of ileal tissues were fixed in 10% neutral buffered formalin, embedded in paraffin, and sectioned. Villus height was measured under a microscope. Based on the upper and lower quartiles of villus height distribution, samples were stratified into a high-villus group and a low-villus group, with 4 biological replicates per group.

Transcriptomic sequencing: Total RNA was extracted from high- and low-villus group samples according to their identifiers. mRNA was enriched using oligo (dT) magnetic beads, reverse-transcribed into cDNA, and sequenced on an Illumina NovaSeq 6000 (Waltham, MA, USA) platform with paired-end 150 bp reads. The RNA sequencing data have been uploaded to (https://doi.org/10.6084/m9.figshare.28760423.v3 (accessed on 1 February 2025).

### 2.3. Alignments and Variant Identification

This study implemented a standardized pipeline to identify reliable RNA editing sites from raw sequencing data. The workflow included the following steps: Raw reads were processed with fastp [23] to remove adapter contamination, low-quality bases (Q-score < 20), and short fragments (<50 bp), generating clean reads for downstream analysis. Clean reads were aligned to the pig reference genome (Sscrofa11.1) using HISAT2 (v0.7.17) [24], producing BAM-formatted alignment files; RNA editing sites were detected using Samtools (v1.9) [25], Picard Tools (v3.1.1) [26], and GATK (v4.0) [27]. All candidate variants were filtered through GATK’s Variant Filtration module with the following criteria: quality by depth (QD) < 2.0; fisher strand bias (FS) > 60.0; MQ rank sum test (MQRankSum) < −12.5; read position rank sum test (ReadPosRankSum) < −8.0; RMS mapping quality (MQ) < 40.0; mean depth must meet inter-individual deviation ≤ 1/3× or ≥3×; strand odds ratio (SOR) > 3.0. Retained RNA editing sites met the following criteria: base quality ≥ 25; total read depth ≥ 5; alternate allele depth ≥ 3; editing frequency between 10 and 100%. The proportion of reads supporting RNA editing relative to total reads at the site was defined as the editing frequency.

### 2.4. Differential Editing Analysis of RNA Editing Events

For each RNA editing site, the number of edited reads in the low-villus-height (LVH) and high-villus-height (HVH) groups was quantified. RNA editing sites were mapped to their corresponding genes in the pig reference genome Sscrofa11.1, and the associated gene was defined as the RNA editing gene. At the gene level, suppose the contribution weight of each RNA editing site is defined equally, and the total edited read counts across all editing sites within a gene are summed to quantify the overall RNA editing level. Differentially RNA-edited genes (REGs) were identified using DESeq2 with thresholds of |log_2_FoldChange| > 1 and *p*-value < 0.05. This analysis highlights genes with significant RNA editing differences between LVH and HVH groups, enabling focused investigation of their functional roles.

### 2.5. RNA Differential Expression Genes Analysis

Gene expression levels were quantified, transcriptomes reconstructed, and TPM (transcripts per million) values calculated using StringTie (v2.2.0) [28]. To identify differentially expressed genes (DEGs), statistical analysis was performed with DESeq2 [29], applying thresholds of *p*-value < 0.05 and |log2FoldChange| ≥ 1. This approach enabled the identification of genes with significant expression differences between the low-villus-height (LVH) and high-villus-height (HVH) groups, thereby highlighting potential functional targets.

### 2.6. Pathway Enrichment

Pathway-based analysis is a powerful approach to elucidate gene functions in complex biological processes. The KEGG (Kyoto Encyclopedia of Genes and Genomes) [30] and Gene Ontology (GO) [31] databases serve as critical public resources for pathway annotations. In this study, gene set enrichment analysis (GSEA) was performed to identify key pathways and biological processes associated with the development of high versus low villus height. Briefly, genes were pre-ranked based on differential expression data (log_2_FoldChange) between the high-villus-height (HVH) and low-villus-height (LVH) groups. GSEA was conducted using the GSEA software (v4.3.3) [32] from the broad institute with default parameters. Enriched pathways with *p* < 0.05 were considered statistically significant.

### 2.7. Correlation Analysis Between Genes RNA Editing and Expression

This study employed integrative multi-omics analysis to explore the potential associations between RNA-edited genes and gene expression levels. Specifically, a nine-quadrant plot was constructed based on the log_2_FoldChange values of differentially edited genes (REGs) and differentially expressed genes (DEGs). The *x*-axis and *y*-axis of the plot represent the log_2_FC values of RNA editing and gene expression, respectively. Threshold lines (|log2FoldChange| ≥ 1) were applied to define significantly different regions. Genes located in high-confidence quadrants (e.g., quadrants 1, 3, 7, and 9), where both RNA editing and gene expression showed significant changes, were prioritized for further investigation. Additionally, the correlation between RNA editing enzymes and villus height was assessed using Pearson’s correlation test in R (v4.2.0) [33], *p* < 0.05 considered statistically significant.

### 2.8. Statistical Analysis

Experimental data were subjected to *t*-test analyses, with a significance threshold set at *p* < 0.05. Graphs were generated using GraphPad Prism 8 software (GraphPad, Santiago, MN, USA). Data were presented as mean ± standard deviation (SD).

## 3. Results

### 3.1. General Characteristics of RNA Editing Events in HVH and LVH

We measured intestinal villus heights in pig samples from our laboratory [34] and observed significant distributional differences among the measured values. Based on the measured villus heights of ileum, we categorized the samples into two groups: the high-villus-height (HVH, N = 4) group (average height: 379.3 μm) and the low-villus-height (LVH, N = 4) group (average height: 290.7 μm) (Figure 1A). Furthermore, analysis of crypt depth—a critical parameter affecting ileal absorption—revealed that the HVH group exhibited significantly greater crypt depths compared to the LVH group (Figure 1B). Additionally, the villus height-to-crypt depth ratio tended to be higher in the HVH group than in the LVH group (Figure 1C).

It is well established that intestinal villus height is closely associated with nutrient absorption capacity. Consequently, elucidating the regulatory mechanisms underlying the differences between high- and low-villus-height groups is of critical importance. By employing an RNA editing analysis pipeline, we revealed the key role of RNA editing in the molecular networks governing villus height regulation (Figure 2). The characteristics of identified RNA editing sites were systematically summarized. Editing types across all samples were classified into distinct categories. C-to-U (C→T or G→A) and A-to-I (A→G or T→A) editing events dominated, collectively comprising ~80% of all detected instances. Less frequent variants, such as A-to-C and A-to-T transitions, highlighted the heterogeneity of the RNA editing repertoire (Figure 2A). Analysis of RNA-editing-induced single nucleotide variants (SNVs) revealed that synonymous substitutions were most abundant, followed by nonsynonymous mutations, which directly impact protein structure and function. Though rare, stop-gain and stop-loss variants exhibited profound effects by altering stop codon positions, potentially disrupting protein integrity and activity (Figure 2B). Spatial distribution analysis showed that the majority of editing events were concentrated in exonic and 3′UTR regions, emphasizing their potential roles in coding sequence regulation. Notably, significant proportions were also observed in upstream, downstream, splicing, and ncRNA intronic regions, suggesting multifaceted regulatory roles for RNA editing across genomic contexts (Figure 2C). To assess RNA editing dynamics, frequency distributions were compared between HVH and LVH groups. Histogram analysis demonstrated substantial variation in editing frequencies across both groups, reflecting baseline differences in RNA editing activity between the two conditions (Figure 2D). A Venn diagram analysis further revealed overlaps and unique components of editing event-associated genes (Figure 2E). Specifically, 14,782 editing events were shared between HVH and LVH groups, indicating that while core editing mechanisms are conserved, distinct regulatory pathways likely operate between the two groups at the RNA edit frequency level.

### 3.2. RNA Editing Genes Are Significantly Enriched in the TNF Signaling Pathway in HVH

To identify differentially expressed RNA editing genes (REGs) between low-villus-height (LVH) and high-villus-height (HVH) groups, we performed differential gene-level analysis. Principal component analysis (PCA) of RNA editing frequencies revealed distinct clustering of HVH and LVH samples, emphasizing clear group separation (Figure 3A). A total of 849 differentially edited genes (*p* < 0.05) were identified between the two groups (LVH vs. HVH), comprising 472 upregulated and 377 downregulated editing genes (Figure 3B). KEGG pathway analysis was performed to investigate the functional roles of REGs. The results revealed that REGs were predominantly enriched in biological processes such as immune system regulation, cell growth and death, signal transduction, and protein folding/sorting/degradation (Figure 3C). Specific pathways included the immune system: IL-17 signaling (ssc04657), T cell receptor signaling (ssc04660), toll-like receptor signaling (ssc04620), and nod-like receptor signaling (ssc04621); cell growth and death: cellular senescence (ssc04218) and cell cycle regulation (ssc04110); signal transduction: TNF signaling (ssc04668) and VEGF signaling (ssc04370); and protein processing: endoplasmic reticulum protein processing (ssc04141) and RNA degradation (ssc03018). Furthermore, gene set enrichment analysis (GSEA) demonstrated that TNF signaling pathway genes exhibited significantly higher editing frequencies in the HVH group (Figure 3D). Functional enrichment analysis suggests that REG genes may exhibit potential associations with biological processes such as the TNF signaling pathway, cell cycle regulation, and growth hormone secretion, indicating possible involvement in maintaining villus height homeostasis. These observational findings provide preliminary insights into the biological roles of REG family genes.

We next examined RNA editing levels in genes related to the TNF signaling pathway via KEGG pathway analysis. We found that these genes (CXCL10: C-X-C motif chemokine 10; MAPK13: mitogen-activated protein kinase 13; TAB2: TGF-beta activated kinase 1; CCL2: C-C motif chemokine ligand 2; CREB3L2: CAMP-responsive element-binding protein 3-like 2; EDN1: Endothelin 1; PIK3R1: Phosphoinositide-3-kinase regulatory subunit 1) exhibited a significantly higher RNA editing level in the high-villus-height (HVH) group compared to the low-villus-height (LVH) group (Figure 4 and Appendix A). This observational finding suggests that enhanced RNA editing regulation in the HVH group may exhibit potential involvement in promoting intestinal villus cell growth, which could be associated with villus height (VH) development.

### 3.3. RNA Expression Genes Are Significantly Enriched in the TNF Signaling Pathway in LVH

Based on the RNA editing analysis described above, we identified a potential association between villus height (VH) in the ileum and the TNF signaling pathway. To further investigate this relationship, we performed RNA expression profiling of ileal tissue samples. A total of 46 differentially expressed genes (DEGs, *p* < 0.05, |Log2FC| ≥ 1) were identified between the low-villus-height (LVH) and high-villus-height (HVH) groups (LVH vs. HVH), including 22 upregulated and 24 downregulated genes (Figure 5A).

KEGG pathway analysis was performed to investigate the functional roles of REGs. The results revealed that DEGs were predominantly enriched in biological processes such as immune system regulation, the digestive system, and the endocrine system (Figure 5B). Specific pathways included the immune system: IL-17 signaling pathway (ssc04657), B cell receptor signaling pathway (ssc04662), Th1 and Th2 cell differentiation (ssc04658), Toll-like receptor signaling pathway (ssc04620), and Th17 cell differentiation (ssc04659); digestive system: fat digestion and absorption (ssc04975), protein digestion and absorption (ssc04974), and vitamin digestion and absorption (ssc04977); and endocrine system: regulation of lipolysis in adipocytes (ssc04923) and PPAR signaling pathway (ssc03320). Based on the KEGG functional enrichment results, differentially expressed genes (DEGs) directly or indirectly regulate immune responses, nutrient digestion/absorption, and inflammatory pathways, contributing to the development of intestinal villus height differences. This observational result suggests that variations in villus height (VH) may be associated with dynamic regulation of specific biological signaling pathways. Notably, both REG and DEG gene sets exhibited enrichment trends in inflammation-related pathways (e.g., TNF signaling pathway). This association implies that RNA editing might indirectly influence villus development through its involvement in inflammatory signaling pathways.

### 3.4. High RNA Editing Attenuates Ileum Inflammation: Integrated Analysis of RNA Editing and Expression

To explore the interplay between the RNA editome and transcriptome in intestinal villus height variation, we employed a nine-quadrant analysis to integrate multi-omics data. Notably, genes such as CXCL10, CCL2, CREB3L2, and PIK3R1 showed significant associations in the ninth quadrant (high expression with low editing efficiency) (Figure 6A). These genes are well-documented in regulatory expressed genes (REGs) for their roles in inflammation and immune responses. Further profiling of seven key inflammation-related genes from REGs revealed that low villus height was associated with elevated expression of pro-inflammatory genes, suggesting that chronic inflammation might impede villus development (Figure 6B). Integrative analysis of regulatory differentially edited genes (REGs) and differentially expressed genes (DEGs) revealed that low-villus samples exhibited elevated expression of inflammation-related genes. However, despite this transcriptional upregulation, RNA-editing-mediated regulatory functions—such as RNA sequence modification, translational control, and protein folding stability—were significantly impaired. This finding suggests that RNA-editing-mediated suppression of inflammatory pathways may play a role in promoting high villus development.

To validate this hypothesis, we examined the correlation between the expression of APOBEC1 (apolipoprotein B mRNA editing enzyme catalytic subunit 1)—a key RNA editing enzyme regulating intestinal RNA editing capacity—and villus height (Figure 6C). Linear regression analysis revealed a statistical association trend (*p* < 0.05), suggesting potential regulatory roles of RNA editing activity in villus height (VH) development. This correlation provides exploratory insights into the biological involvement of RNA editing in intestinal morphogenesis.

## 4. Discussion

This study focused on the ileal villus height variation in crossbred pigs (Landrace × Yorkshire × Duroc, DLY) by dividing them into a high-villus group and a low-villus group. A set of inflammatory signaling pathways regulating ileal villus height were identified.

Transcriptomic analysis revealed 22 upregulated and 24 downregulated differentially expressed genes (DEGs) between the two groups. Notably, the limited number of DEGs aligns with our previous findings in the jejunum of DLY pigs, where similar patterns of villus-height-associated gene expression were observed [34]. These results suggest that during intestinal development in DLY pigs, gene regulatory networks may exhibit high stability, and the phenotypic differences in villus height could be driven by the synergistic effects of a small subset of key regulatory genes. Therefore, investigating the limited number of key genes linking villus height phenotypic differences is of critical importance. Notably, these REGs are predominantly enriched in the IL-17 and TNF signaling pathways—two classical pro-inflammatory cascades. Previous studies have demonstrated that hesperidin suppresses intestinal damage by inhibiting inflammatory mediators such as IL-17A, TRAF6, phosphorylated p38 (P-p38), and AP-1 in the colons of pigs, thereby enhancing mucosal barrier integrity. This evidence highlights the therapeutic potential of targeting the IL-17 pathway through hesperidin supplementation to improve gut health [35]. Furthermore, another study found that heat stress induces a systemic inflammatory response in pigs, whereas environmental cooling interventions can significantly increase this inflammatory state. During this process, jejunal villus height is markedly reduced, accompanied by a significant increase in TNF-α levels, further underscoring the central role of the TNF signaling pathway in inflammation regulation [36,37]. Consistently, this study observed that differentially expressed inflammatory genes (e.g., CXCL10, CCL2, CREB3L2, and PIK3R1) were significantly upregulated in the low-villus group. These genes impair intestinal health by triggering excessive inflammatory responses, ultimately leading to the development of reduced villus height. Functional enrichment analysis reveals that REG genes exhibit enrichment trends in biological processes related to intestinal epithelial cell proliferation, particularly cell cycle regulation. Given the established link between cell cycle dynamics and villus height (VH) development, this observational association suggests that REG may indirectly influence epithelial cell proliferation through modulating RNA editing events in cell cycle-related genes, potentially contributing to villus morphogenesis.

During intestinal development, classical chemokine genes such as CXCL10 and CCL2 are widely recognized as biomarkers of intestinal inflammatory injury. Numerous studies have demonstrated that reducing the expression of these chemokines offers effective strategies for alleviating intestinal inflammation [38,39]. For instance, Lactobacillus jensenii TL2937 significantly suppresses acute intestinal inflammation by downregulating CXCL10 and CCL2 expression [40]. Another approach involves targeting upstream regulators of CCL2: F-box and WD repeat domain-containing protein 7 (FBXW7), an E3 ubiquitin ligase in the gut, promotes Ccl2 and Ccl7 expression, exacerbating inflammatory bowel diseases (IBDs). Research has shown that inhibiting FBXW7 activity reduces CCL2 expression and limits the recruitment of pro-inflammatory macrophages (e.g., M1 macrophages), thereby mitigating intestinal inflammation [41,42]. In this study, a consistent phenomenon was observed: the inflammatory chemokines CCL2 and CXCL10 were significantly upregulated in the low-villus-height ileal group, potentially impairing intestinal development through chronic inflammatory responses. Notably, at the post-transcriptional level, we first identified that CCL2 and CXCL10 exhibited markedly enhanced RNA editing in the high-villus-height ileal group. Specifically, there are two mechanisms by which RNA edits may disrupt their function: codon recoding, where A→I (adenosine-to-inosine) and C→U (cytidine-to-uridine) editing converted codons encoding critical amino acids into stop codons (e.g., UAG and UGA), leading to premature termination of protein translation, and RNA secondary structure alteration, in which editing-induced base-pairing changes near editing sites form aberrant stem-loop structures, hindering ribosome binding or mRNA stability, thereby suppressing translational efficiency. This RNA-editing-mediated functional suppression significantly reduced the pro-inflammatory effects of CCL2 and CXCL10, limiting immune cell infiltration and barrier damage, ultimately promoting healthy villus development. In summary, at the gene expression level, high CCL2 and CXCL10 expression may exacerbate villus atrophy through inflammation; however, at the RNA editing level, higher editing levels effectively block chemokine function, thereby protecting intestinal villus integrity.

The regulatory role of RNA editing in inflammation has garnered increasing attention [43]. Since RNA editing is mediated by specific editing enzymes (e.g., ADAR and APOBEC families), numerous studies have explored RNA editing guide systems or enzyme-targeted interventions (e.g., CRISPR/CasRx technology and deaminase inhibitors) to investigate their therapeutic potential in inflammatory diseases [44,45,46,47]. Some studies have shown that the intestinal-specific expression of the RNA editing enzyme APOBEC1 is closely linked to lipid absorption in the gut. Compared to Apobec1^(−/−)^ global knockout mice, Apobec1(Int/Int) mice with intestinal-specific APOBEC1 expression generate smaller APOB48-containing chylomicrons through RNA editing [48]. Even when the mice are fed a high-fat diet, adenovirus-mediated expression of the ApoB mRNA editing enzyme effectively reduces plasma LDL cholesterol and esterified cholesterol levels in LDLR^−/−^ (low-density lipoprotein receptor-deficient) mice [49]. Furthermore, emerging evidence highlights APOBEC1’s role in inflammation: its RNA editing activity modulates the abundance of proteins encoded by inflammatory genes, thereby coordinating multiple cellular pathways. For instance, mice lacking APOBEC1-mediated editing in microglia exhibit elevated expression of pro-inflammatory cytokines (e.g., IL-6, IL-1β, and TNF-α), lysosomal dysfunction, and ultimately compromised systemic health [50]. Our study identified a consistent phenomenon: on one hand, APOBEC1 gene expression showed a significant link with ileal villus height (R = 0.81, *p* < 0.05); on the other hand, in the group with low APOBEC1 expression and reduced villus height, RNA editing levels of inflammatory chemokines CCL2 and CXCL10 were decreased, while their mRNA expression levels were significantly elevated. These findings suggest that during ileal development in crossbred pigs, endogenous APOBEC1 enhances RNA editing capacity of TNF signaling pathway-related genes, thereby suppressing pro-inflammatory cytokines (e.g., CCL2 and CXCL10) and establishing an anti-inflammatory balance to promote intestinal villus growth and maintain gut health. Current research on APOBEC1 expression variation primarily focuses on environmental stressors, inter-individual genetic heterogeneity, and housing condition differences. In this study, using a homogeneous pig cohort (controlled genetic background, standardized housing environment, and uniform diet), the observed APOBEC1 expression heterogeneity may exhibit potential associations with gut microbiota composition and its metabolite dynamics. This microbe–host interaction pattern could indirectly modulate APOBEC1 expression through intestinal epithelial homeostasis disruption (e.g., short-chain fatty acid-mediated epigenetic regulation), though the underlying molecular mechanisms require further validation via microbiota transplantation experiments.

This study is limited by a small sample size (n = 4), which may compromise statistical power and affect the reliability of differential gene detection (e.g., DESeq2 algorithm based on negative binomial distribution is sensitive to sample size). To mitigate false-positive risks, we applied stringent criteria (*p* < 0.05, |log2FC| ≥ 1) combined with functional enrichment analysis for biological relevance validation. Despite these limitations, our findings provide exploratory insights into RNA editing regulatory networks associated with intestinal epithelial development and offer potential targets for villus height (VH) homeostasis research.

## 5. Conclusions

This study, based on ileal tissues with varying villus heights (VHs), observed potential associations between inflammatory regulatory networks and intestinal villus development. Functional analysis suggested elevated expression levels (*p* < 0.05) of TNF signaling pathway genes in the low-VH group, which may correlate with enhanced inflammatory status in intestinal epithelial cells. Concurrently, reduced RNA editing events in this pathway exhibited hypomodification trends, potentially affecting gene expression stability. Our findings provide exploratory insights into TNF-mediated inflammatory regulation during gut development and highlight RNA editing as a potential endogenous anti-inflammatory regulatory mechanism. These results offer theoretical references for RNA editing-targeted interventions in intestinal inflammation.

## Data Availability

https://doi.org/10.6084/m9.figshare.28760423.v3, accessed on 1 February 2025.

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
