# Peer review of "APOBEC1-Dependent RNA Eiting of TNF Signaling Orchestrates Ileal Villus Morphogenesis in Pigs: Integrative Transcriptomic and Editomic Insights"

_animals, 2025, doi:10.3390/ani15162419_

Round 1

Reviewer 1 Report

Comments and Suggestions for Authors

Review of the article «APOBEC1-dependent RNA editing of TNF signaling orchestrates ileal villus morphogenesis in pigs: integrative transcriptomic and editomic insights»

This study presents a well-designed investigation that offers a novel perspective on gut health regulation. The authors propose and substantiate the hypothesis that RNA editing is a key mechanism in controlling inflammation and villus morphogenesis, moving beyond classical gene expression analysis. This approach may open new avenues for understanding the molecular foundations of intestinal homeostasis.

Despite the high quality of the work, several aspects require clarification or could be strengthened to make the conclusions more robust.

1) Sample Size Justification. The most apparent limitation is the small sample size (n=4 per group). Although the selection of extreme phenotypes (upper and lower quartiles) enhances the biological signal—likely contributing to statistically significant results—such a small group is sensitive to individual variation. The authors should explicitly acknowledge this limitation in the "Discussion" section. It is recommended that they emphasize that, despite the statistical significance, the findings should be considered preliminary and require validation in a larger cohort. This will enhance the objectivity and credibility of the study.

2) Correlation and Causation. The study reports a strong positive correlation (R=0.81) between APOBEC1 expression and villus height (Figure 6C). However, the study design does not allow for definitive conclusions regarding causality. The authors are advised to soften their language by replacing causal terms (e.g., “drives,” “is essential for”) with more appropriate associative terms (e.g., “is associated with,” “correlates with”). Additionally, the “Discussion” could benefit from a brief consideration of alternative scenarios—for example, the possibility that both APOBEC1 expression and villus morphology are regulated by a common upstream factor.

3) Mechanism of Gene Function Suppression. The central concept of the manuscript is that RNA editing suppresses the function of pro-inflammatory genes such as CXCL10 and CCL2. However, the article lacks detailed data on the specific editing sites in these genes and their functional consequences. To reinforce this crucial point, the authors should provide additional information about the nature of these edits. Do the identified sites result in nonsense mutations (e.g., premature stop codons), missense mutations (amino acid substitutions in functional domains), or are they located in untranslated regions (e.g., 3'UTR) that could affect mRNA stability?

4) Quantitative Assessment of Gene Editing Level. The approach of summing all edited reads to estimate the overall RNA editing level of a gene (REG) is a pragmatic solution. However, it assumes that all editing sites contribute equally to gene function, which may not be the case. It would be helpful for the authors to briefly justify this approach in the “Materials and Methods” section and acknowledge its underlying assumptions.

This work opens many directions for future research, raising several questions that would be interesting to address in the "Discussion" section:

5) Search for APOBEC1 Regulators. What regulates APOBEC1 itself? If this enzyme is indeed critical, what controls its intestinal expression? Could environmental factors such as microbiota composition, dietary components (e.g., short-chain fatty acids, polyphenols), or stress influence APOBEC1 expression and thereby modulate the intestinal "editome"?

6) Investigation of Other Genes. The study identified 849 differentially edited genes. While the focus was placed on genes within the TNF signaling pathway, what about the rest? What other biological processes—beyond inflammation—might be fine-tuned through RNA editing in the intestine? An expanded analysis of the REG list may uncover parallel or complementary regulatory pathways involved in nutrient absorption, epithelial proliferation, or barrier function.

This is a  study that lays the foundation for a new understanding of intestinal biology. Addressing the points outlined above would enable a deeper exploration of fundamental mechanisms and support the future development of innovative strategies for promoting animal health.

Author Response

Dear reviewer, through literature review and discussion, and in response to your valuable suggestions, we have made improvements and revisions. The details of the modified content will be referred to in the attachment. Please pay attention to check it.

Reviewer 2 Report

Comments and Suggestions for Authors

The manuscript describes the differences in REGs and DEGs between high and low villus exhibiting  group's ileum samples of crossbred swines. It is an interesting concept and the experimental setup is adequate the methods are and up-to-date. The results are presented at high quality. At some point the interpreation and the conclusions are not clear, which could be improved. Fig 6 B shows seven genes expression, out of which three are not significantly different between the HVH and LVH groups. In spite of it the sentence in lines 317-320 describes a strong, generalized (over)statement about that chronic inflammation might impede villus development. This is more precisely described in the  conclusion lines 369-371. Other parts of the conclusion are repetitive e.g. 389-395. 

Author Response

(The authors gave the same response as above.)

Round 2

Reviewer 1 Report

Comments and Suggestions for Authors

 Accept in present form